# Mechanical Properties of Abaca–Glass Fiber Composites Fabricated by Vacuum-Assisted Resin Transfer Method

**DOI:** 10.3390/polym13162719

**Published:** 2021-08-13

**Authors:** Marissa A. Paglicawan, Carlo S. Emolaga, Johanna Marie B. Sudayon, Kenneth B. Tria

**Affiliations:** Department of Science and Technology, Industrial Technology Development Institute, Bicutan, Taguig 1631, Philippines; csemolaga@itdi.dost.gov.ph (C.S.E.); johanna.sudayon@yahoo.com (J.M.B.S.); tria_kenn@yahoo.com (K.B.T.)

**Keywords:** abaca fiber, hybrid, glass fiber, VARTM, mechanical properties

## Abstract

The application of natural fiber-reinforced composites is gaining interest in the automotive, aerospace, construction, and marine fields due to its advantages of being environmentally friendly and lightweight, having a low cost, and having a lower energy consumption during production. The incorporation of natural fibers with glass fiber hybrid composites may lead to some engineering and industrial applications. In this study, abaca/glass fiber composites were prepared using the vacuum-assisted resin transfer method (VARTM). The effect of different lamination stacking sequences of abaca–glass fibers on the tensile, flexural, and impact properties was evaluated. The morphological failure behavior of the fractured-tensile property was evaluated by 3D X-ray Computed Tomography and Scanning Electron Microscopy (SEM). The results of mechanical properties were mainly dependent on the volume fraction of abaca fibers, glass fibers, and the arrangement of stacking sequences in the laminates. The higher volume fraction of abaca fiber resulted in a decrease in mechanical properties causing fiber fracture, resin cracking, and fiber pullout due to poor bonding between the fibers and the matrix. The addition of glass woven roving in the composites increased the mechanical properties despite the occurrence of severe delamination between the abaca–strand mat glass fiber.

## 1. Introduction

The market for natural fiber-reinforced polymer composites (NFRPCs) has been showing an increasing trend in recent years for automotive, aerospace, construction, and sports applications. The use of NFRPCs offers the advantages of being environmentally friendly and lightweight, having a low cost, and having a lower energy consumption during production. The main disadvantage of natural fiber as a reinforcement for composites is the poor compatibility of the fibers and the matrix due to the hydrophilic nature of natural fibers, which leads to poor bonding at fiber/matrix interfaces. One effective approach to improving the properties of natural fiber composites is hybridization with synthetic fibers, which can tailor the properties to the desired application. Reinforcement by two or more fibers in a single matrix may lead to hybrid composites with a great range of material properties [1]. The hybrid composites are very dependent on the reinforcing fibers used in the matrix and possess different advantages and disadvantages [2]. The hybrid composite has the advantage of overcoming the disadvantage of natural fibers used in the composite [3,4]. Glass fiber is a common and versatile synthetic fiber in reinforcing thermosets and thermoplastics. They exhibit useful bulk properties such as hardness, transparency, resistance to chemical attack, stability, and inertness as well as desirable fiber properties such as strength, flexibility, and stiffness [5,6]. Several researchers have tried using hybrids of two or more fibers, either both natural fibers or a combination of natural fibers and synthetic fibers with thermosets and thermoplastics. The effect of glass fiber hybridization with natural fibers on the mechanical and thermal properties of thermoset composites was previously reviewed [7,8,9,10]. Some specific natural fiber hybrids with glass fiber were also reported: jute/glass in unsaturated polyester composites [11,12], jute/glass in epoxy [13,14,15,16], kapok/glass–unsaturated polyester [17,18], sisal/glass–unsaturated polyester [19,20,21,22], bamboo/glass–unsaturated polyester [23], pineapple leaf/glass–polyester [24], flax and glass fiber [25], and the effect of the stacking sequence of cotton/glass fiber–polyester [26].

One of the natural fibers is abaca fibers, known as Manila Hemp. Abaca fiber is obtained from the stalk of the *Musa Textilis Nee*. It is composed of 53–63% cellulose, 7–9% lignin, and 20–25% hemicellulose [27,28]. The Philippines is the largest producer of abaca fiber, supplying it to about 87% of the world for the production of cordage, specialty papers, textiles, furniture and fixtures, handicrafts, novelty items, meat casing, cosmetics and skin care products, composites for automotive, construction, and other industrial applications [29]. Abaca fibers have high tensile strength and impact strength that can be used for automotive applications. Several researchers have studied the properties of abaca fiber-reinforced thermoplastics [30,31,32,33] and thermoset composites [34,35,36,37,38,39]. The hybridization of abaca fibers with glass fibers and other natural fibers has also been studied [40].

Vacuum-assisted resin transfer molding (VARTM) technology is ideal for the fabrication of large-scale composites using synthetic fibers. The process offers safer, clean, and economical manufacturing of the composite that reduces voids, impurities, and bubbles and provides a homogeneous material with improved mechanical properties. There are few research studies on the hybridization of abaca fiber with glass fiber and manufacturing of natural fiber-reinforced composites using VARTM technology [34,35,41].

This study aimed to determine the effect of lamination sequence of the hybridization of an abaca–glass fiber unsaturated polyester composite and to demonstrate the fabrication of composites using the vacuum-assisted resin transfer manufacturing (VARTM) method for marine application. The mechanical properties in terms of tensile properties, flexural properties, and impact strength of the hybrid abaca–glass fibers were evaluated. An analysis of the failure behavior of abaca–glass fiber using the VARTM process was performed using Scanning Electron Microscopy (SEM) and X-ray Computed Tomography.

## 2. Materials and Methods

### 2.1. Materials

S-2 (Streaky Two)-grade abaca fibers in the form of plain fabrics in the 0/90 direction containing 19 fabrics in warp and 11 fabrics in the welf direction were used in this study, similar to that in another paper [33]. This grade is produced from the inner, middle, and next-to-outer sheaths of the abaca stalk. The S-2 grade variety is an excellent quality of abaca based on the strength, cleaning, color, texture, and length of the fiber, which was taken from the Bicol Region of the Philippines. The glass fibers used in this study are in the form of chopped strand mat (CSM) CSM300 (300 g/m^2^) and CSM450 (450 g/m^2^), and woven roving (WR), 600 g/m^2^ (Polymer Products, Pasig, Philippines). The glass fibers in the chopped strand mat form were laid randomly across each other and were held together by a binder, while those in woven roving were made into a bidirectional fabric. The matrix used was isophthalic polyester resin, with cobalt naphthenate as the accelerator and methyl ethyl ketone peroxide as the hardener. Isophthalic polyester resins are better suited for corrosion environments and elevated service temperatures and have greater mechanical properties than their orthophthalic counterpart. These chemicals were purchased from Polymer Products Inc., Pasig, Philippines.

### 2.2. Composite Lamination Preparation

VARTM has been used in many applications because of its capability of fabricating composites with good quality within a shorter time and at a relatively low cost. Nine different stacking lamination sequence samples were fabricated using the VARTM process, as shown in Table 1. The basic steps in the VARTM fabrication include the following: (1) mold preparation, which involves cleaning of the mold surface using ethyl alcohol and applying poly(vinyl alcohol) (PVA) as a release agent on the surface of the mold; (2) fabric lamination, in which the lamination stacking sequence was fixed in a mold with dimensions of 30 cm × 30 cm; (3) sealing the mold and creating a vacuum, wherein multiple layers of fabrics were covered with vacuum bags and tightly sealed with silicone tape; (4) resin transfer, which allows the resin with the catalyst to fill the contents of the mold using an injection pressure of 25 psi (0.17 MPa) using vacuum pump controller, Ulvac-CLD-051 (Ulvac Kiko Inc, Shanghai, China); and (5) curing the sample, which involves pre-curing of the sample in the mold for 4 to 5 h followed by removal of the sample from the mold and storing for 24 h at room temperature. The fiber volume fraction is calculated from Equations (1) and (2).
Vf = (*vf*/*vm*)(1)
Vf + Vm = 1(2)
where Vf = fiber volume fraction; Vm = matrix volume fraction; *vf* and *vm* = volume of fiber and matrix, respectively.

### 2.3. Characterization

The tensile tests after conditioning at 25 °C were carried out on a Universal Testing Machine (UTM) model Instron 5585H (Instron, Norwood, MA, USA) equipped with a 10 kN capacity load cell. The tests were performed in accordance with ASTM standards D3039. The crosshead speed used was 5 mm/min for tensile testing. All of the samples were cut in the welf direction. The average values of the mechanical properties were obtained from 6 specimens. The flexural test procedures and values were obtained in accordance with ASTM D790 and ISO 178 using Shimadzu Universal Testing Machine (AGS-50kNXD, Shimadzu, Tokyo, Japan) with specimens of nominal dimensions of 50 mm × 25 mm × 2.5 mm, a span length of 90 mm, and a crosshead speed of 0.71 mm/min. The izod impact test was performed in accordance with ASTM D256 using a Zwick/Roell 5.5P pendulum (Zwick Roell Group, Ulm, Germany) with a nominal work capacity of 1 and a theoretical impact velocity of 3.458 m/s. The dimensions of the specimen are 63.5 mm × 12.7 mm × 3 mm.

The tensile-fractured specimens were scanned on 3D X-ray Computed Tomograms (North Star Imaging, Inc., Rogers, MN, USA). The system was equipped with an X-ray source of microfocus tube type with a minimum focal spot size of 42 µm. The energy beam used was 70 kV, and the distance from the specimen and the detector was set to 144.219 mm to provide a degree of phase contrast to facilitate the visualization of crack and delamination features. The data acquisition was carried out with an exposure duration of 20 m 40 s and no pre-filtration. The number of projections was set to 4000 projections, and the average number of frames was 2. The NSI efX-CT (North Star Imaging, Rogers, MN, USA) was used for data visualization and reconstruction of the sample.

The morphology of the composites and interfacial bonding between the fiber and the unsaturated polyester matrix was examined using a Field Emission Scanning Electron Microscope (Helios Nanolab 600, FEI, Hillsboro, OR, USA) to study the changes in the tensile-fractured surfaces of the composites. The samples were taken at magnification of 500×. The samples were sputter-coated with gold for 40 s.

## 3. Results and Discussion

### 3.1. Mechanical Properties

#### 3.1.1. Tensile Properties

Figure 1 shows the tensile strengths of different lamination stacking sequences, which are very dependent on the tensile properties of each fiber component. Similar results were obtained by Sinha [39]. For instance, the laminates (coded 1 and 2) containing only glass fiber reinforcement with different CSM and WR showed that the maximum tensile strength values range from 287 to 298 MPa and a tensile modulus of 17.7 GPa, respectively. The change in lamination stacking sequence and volume fraction of the fibers resulted in different tensile strengths. The tensile strength and tensile moduli of laminates coded 1 and 2 were almost the same despite the difference in the arrangement of CSM and WR. On the other hand, if the hybrid composites such as in laminate coded 3 contain seven layers of abaca fibers and CSM 450, the tensile strength resulted in 79 MPa and a tensile modulus of 7.06 GPa. The addition of CSM 300 (laminate coded 4) did not contribute much to the strength of the composite, which gave only a tensile strength of 95.1 MPa but increased in stiffness to 20.5 GPa. These results were lower than the abaca fiber-reinforced composite previously reported [34,35]. Replacing the CSM 450 with WR as in the case of laminate coded 5, the resulting tensile properties were increased. On the other hand, when the number of layers of abaca fibers was reduced from 7 to 3 but the same lamination stacking sequence was maintained, the tensile strength of laminate coded 6 increased to 143 MPa, but the tensile modulus decreased from 19.5 GPa to 9.08 GPa. The laminate coded 7, consisting of layers of CSM 300 and 450 followed by 3 layers of abaca fibers and WR and then additional layers of abaca fibers, resulted in a lower tensile strength of 101 MPa and a tensile modulus of 11.1 GPa. The results were almost the same in laminate coded 4. It seems from the results that the additional layer of WR did not give strength to the composites. Following the same stacking sequence of laminate coded 8, by just reducing the layers of abaca fibers, the tensile strength and modulus were 128 MPa and 13.2 GPa, respectively. However, when the arrangement of abaca was changed by sandwiching the three layers of abaca fiber with layers of CSM 300 and CSM 450 followed by WR and CSM 300, as in the case of laminate coded 9, the tensile strength was improved, resulting in 135 MPa and a tensile modulus of 12.22 GPa. The tensile properties of the hybrid composites were enhanced with a decreasing amount of abaca fibers. However, decreasing the volume fraction of abaca fibers and additional layer of WR increased the tensile properties of the laminates due to the higher volume fraction of glass fibers. The results showed some similarity to the work of Zhang et al. [25] that reported an improvement in the tensile strength of flax-glass fiber hybrid composite with increase in percent fiber volume in hybrid composites. Ramnath et al. [42] reported that the tensile strength and shear strength of jute–abaca hybrid composites were better than the abaca fiber alone. Thus, the tensile properties in this study were mainly influenced by the volume fraction of abaca fibers, glass fibers, and the arrangement of stacking sequences in the laminates. Among the hybrid composites, laminates coded 5, 6, 8, and 9 show tensile strengths of 30 MPa and above. Laminates coded 4 and 5, on the other hand, showed the highest tensile modulus. This shows that laminate coded 5 exhibited the best combination of high tensile strength and tensile modulus.

#### 3.1.2. Flexural Properties

Figure 2 shows the flexural properties of the hybrid composites made up of glass fibers and abaca fibers. The laminates with code 1 and 2 are made of glass fibers only with different layers of CSM and WR. Similar to the results of tensile properties, the laminates coded 1 and 2 have almost the same flexural strengths and flexural moduli: 424 and 433 MPa, and 13.9 and 17.8 GPa, respectively. The hybrid of different lamination sequences resulted in lower flexural strength, specifically with a higher number of layers of abaca fibers. For instance, the addition of seven layers of abaca fiber in CSM 450 (laminate coded 3) tends to decrease the flexural strength to 106 MPa and flexural modulus to 5.36 GPa. The decrease in flexural properties maybe attributed to low fiber volume fraction (Table 1) and poor interfacial interaction of abaca fibers and the matrix, which is discussed in detail in section on the failure behavior of the composites. However, an increase in flexural strength of 235 MPa and flexural modulus of 8.59 GPa with an additional layer of CSM 300 for laminate coded 4 was observed. The stiffness further increased when the CSM was replaced with WR in laminate coded 5, resulting in a flexural strength of 224 MPa and a flexural modulus of 9.75 GPa. It was observed that, when the layer of abaca fiber was decreased to three (laminate coded 6), the flexural properties increased to 300 MPa and 10.4 GPa. The laminate coded 7 exhibited lower flexural properties. In contrast, the laminates coded 8 and 9 resulted in 306 MPa and 298 MPa for flexural strength and 7.15 GPa and 9.29 GPa flexural modulus. It was found that increasing the percent volume of glass fibers also increases the flexural properties of the composites. Similar findings were observed by Prabhakaran et al. [43], in which the flexural modulus of the hybrid composites was increased when the flax fibers were replaced by glass fibers. The flexural properties were generally controlled by the outer layers of the laminate composites. A similar behavior was reported by using jute/glass fiber laminates [20] and cotton/glass fiber laminates [26]. The laminates with the highest flexural strengths are laminates coded 6, 8, and 9, while those with the highest flexural moduli are laminates coded 5 and 6. In terms of flexural modulus; therefore, the laminate with the best combination of strength and modulus is laminate coded 6.

#### 3.1.3. Impact Strength

The lamination sequence and components have significant effects on the impact strength, as shown in Figure 3. The laminates coded 1 and 2, showing different stacking sequences of glass fiber, have similar impact strengths. An abrupt decrease in impact strength was observed when seven layers of abaca were added to CSM 450 (laminate coded 3), although a 25% increase was seen with the addition of strand mat 300 (laminate coded 4). The increase observed may be attributed to the increase in fiber volume fraction. However, when the CSM 450 was replaced with WR, an improvement in impact strength of 83% was attained (laminate coded 5). These results were similar to those of Portella [26] and de Rosa [44], indicating that, when the glass fiber layers were located away from the central axis, it resulted in a higher impact strength. Reducing the layers of abaca fiber (laminate coded 6) produces a minimal increase. In these particular laminates, the number of layers of abaca fiber has less significant effect on impact strength, which is not always the case. In particular, laminates coded 7 and 8 containing same lamination sequence but a different number of abaca fiber layers exhibited a 42% improvement in impact strength with a lower abaca fiber content. The laminates coded 7 and 9 have a similar amount of abaca layers and have low impact strength. Of the hybrid composites, laminates coded 5, 6, and 8 show the highest impact strengths. This indicates that good impact strength can be achieved with abaca fibers sandwiched between glass fibers.

### 3.2. Failure Morphology of the Laminates

The effect of the addition of abaca fibers and the results of tensile properties of different laminates can be further explained by their failure behavior. Figure 4 shows digital photos of the tensile fractures of all composite laminates. The laminates coded 1 and 2 comprising only glass fibers exhibited different fracture behaviors depending on the laying sequence. The brittle fractures of CSM and matrix occurred at the same time, and some fiber breakage and delamination of WR during the tension loading process were observed in laminate coded 1, as shown in Figure 4a,b. The behavior can be clearly observed in different areas of the cross section of the laminate coded 1 using X-ray Computed Tomography, as shown in Figure 5a. A similar performance was observed when abaca fiber was added to CSM for laminates coded 3 and 4. In the X-ray CT of Figure 5b, the cross section of the fractured samples of laminate coded 4 displayed the presence of voids. The layers of CSM and abaca in the laminates coded 5 and 6 showed brittle fractures, as seen in digital photos (Figure 4a,b), which is similar to that of laminates coded 3 and 4 with some delamination of WR. To give a better understanding on the failure behavior of the hybrid laminates, the cross section of an unfractured tensile specimen of laminate coded 6 was also viewed under X-ray CT for comparison with the tensile fractures, as shown in Figure 5c. It can be seen that some voids are also visible in the unfractured laminate coded 6. On the other hand, the X-ray CT image of the cross section of the fractured sample of laminate coded 6 is shown in Figure 5d. As can be seen in the image, clear damage in the area of CSM with fiber breakage of WR was observed. Digital images of laminates code 7, 8, and 9 are shown in Figure 4a–c. The cross section in Figure 4c where the matrix and fiber contain CSM and abaca fibers were completely damaged while those areas containing WR exhibited severe delamination. A clear image of the laminates coded 8 and 9 is shown in Figure 5e,f. It was further observed that a brittle fracture occurred in the laminates with abaca fibers. This behavior is prominent in the laminates coded 3 and 4, where it contains more layers of abaca fibers. Due to the smaller volume fraction of abaca fibers and more layers of glass fibers in the laminates coded 6 to 9, brittle fracture was not observed.

The results of the tensile properties can be further explained by the surface failure morphology. SEM images are shown in Figure 6. The higher results of the tensile strength of pure glass fibers composites (laminate coded 1) may be due to the strong bond between the fiber and the matrix, and to the higher fiber volume fraction, as shown in SEM image (Figure 6a). However, when the glass fibers were hybridized with abaca fibers, poor interfacial interaction of the abaca fibers, the glass fibers, and the matrix was observed. This behavior is prominent in the laminate coded 3, which contains seven layers of abaca fibers. Matrix cracking, abaca fiber pull-out (see arrow), and poor interfacial between the abaca fiber and unsaturated polymer were seen in this laminate, as shown in the circle (Figure 6b). It was further observed that the abaca fibers absorbed more resin than the glass fibers for all of the hybrid composites containing abaca fibers; thus, the composites consist of a low fiber volume fraction. The poor wettability of abaca could also be observed in the vicinity of interfaces between the fibers and matrix (see circle) and matrix cracking (see circle) of Figure 6c. In laminate coded 5, good adhesion of glass fibers and the matrix can be clearly seen in Figure 6d, while the location of the abaca fibers created severe voids and delamination due to fiber pull-out, as indicated by the arrow. A similar behavior was also seen in Figure 6e. Due to severe delamination of WR in laminate coded 8, only part of the CSM and part of the abaca fibers can be seen in Figure 6f. Figure 6g shows an additional proof that abaca fiber has poor adhesion where voids are prominent (as shown in circles). Based on the failure morphology analysis of the hybrid composites, laminates coded 5 and 6 show the best properties. These laminates pertain to hybrid composites having seven and three layers of abaca sandwiched between the layers of glass fibers.

## 4. Conclusions

In this paper, different stacking sequences of hybrid abaca–glass fiber composites were prepared using the VARTM method. The mechanical properties of the hybrid composites were mainly dependent on the volume fraction of abaca fibers and glass fibers, and the arrangement of stacking sequence in the laminates. It was found that the tensile strength, flexural strength, flexural modulus, and impact strength of laminates increase with increasing amounts of glass fiber. Generally, the flexural properties were controlled by the outer layers of the laminate composites. The hybrid composites containing strand mat glass fibers and abaca fibers resulted in brittle fracture of fibers, resin cracking, fiber pullout, and voids, while delamination occurred in the composites with woven roving. This failure may be due to the high interlaminar stresses that are usually connected to the lowest through-thickness strength. Based on the mechanical properties and failure morphology analysis of the glass fiber–abaca composites, laminates coded 5 and 6 show the best properties for the hull of the boat and other industrial applications.

## 5. Patents

Two applications of utility model were filed at the Intellectual Property Office of the Philippines under registration numbers UM-2-2021/050303 and UM-2-2021/050307.

## Figures and Tables

**Figure 1 polymers-13-02719-f001:**
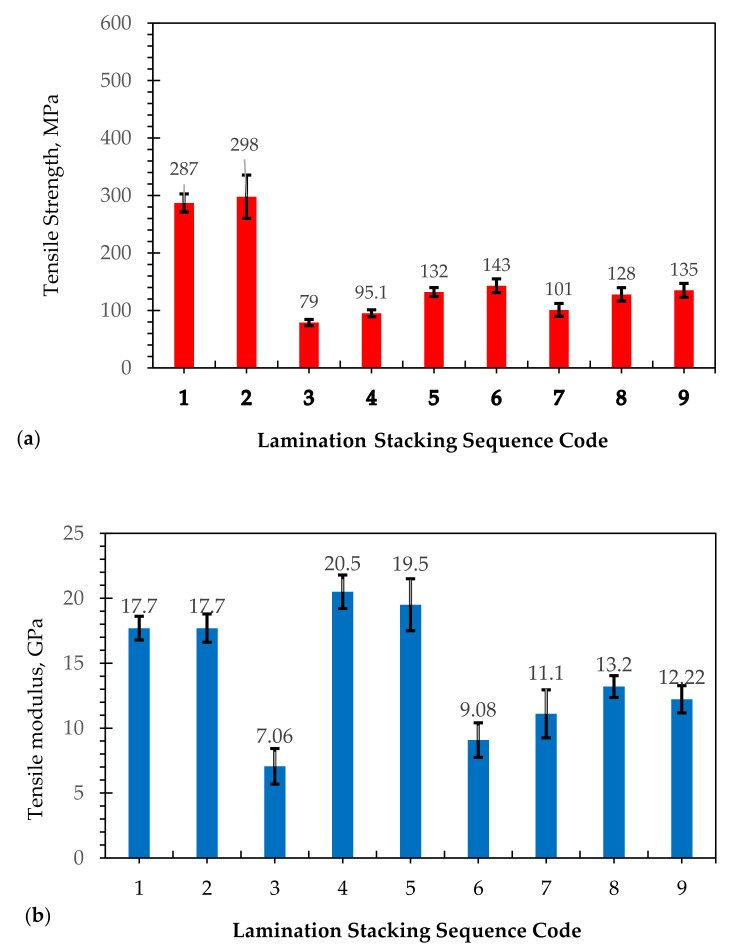
(**a**) Tensile strength and (**b**) modulus of elasticity of different lamination stacking sequences of hybridization of abaca/glass fiber fabricated by VARTM. Note: The numbers 1-9 pertain to lamination stacking sequence code (see Table 1).

**Figure 2 polymers-13-02719-f002:**
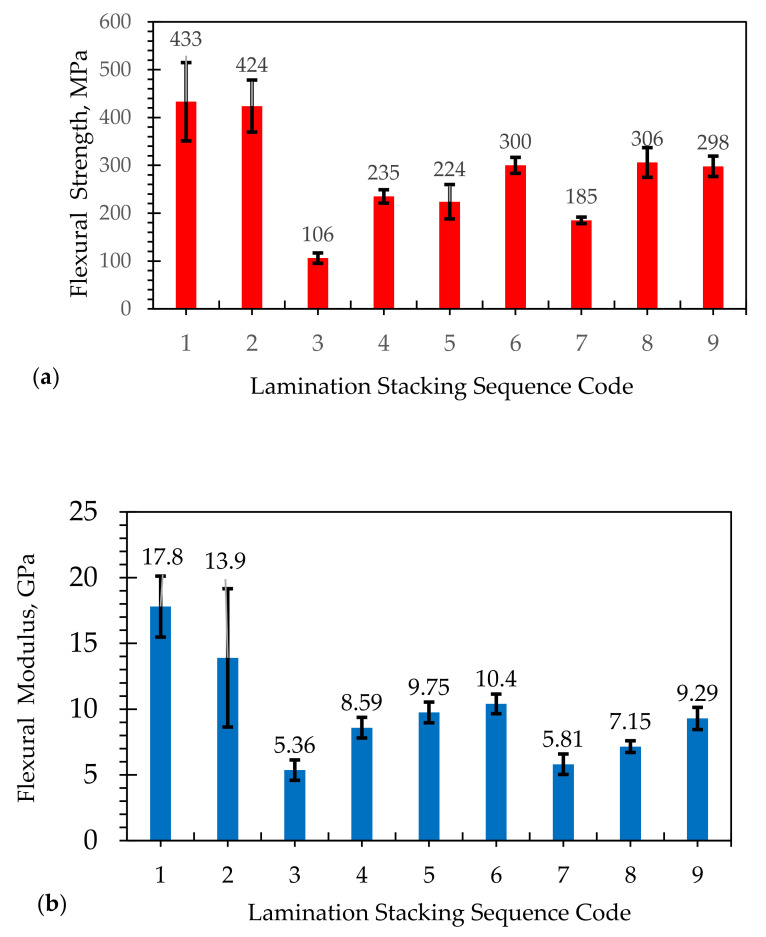
(**a**) Flexural strength and (**b**) flexural modulus of elasticity for different lamination stacking sequences of hybridization of abaca/glass fiber fabricated by VARTM. Note: The numbers 1–9 pertain to lamination stacking sequence code (see Table 1).

**Figure 3 polymers-13-02719-f003:**
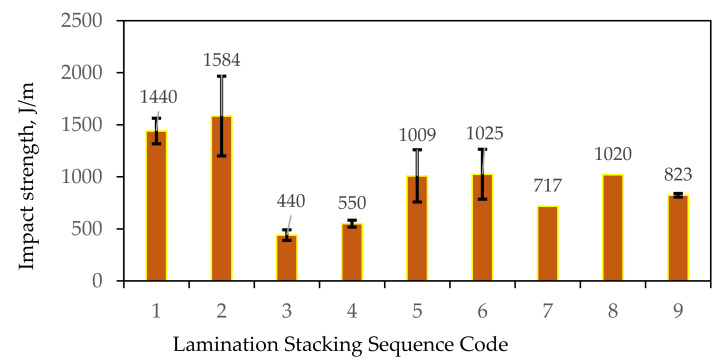
Impact strength of different lamination samples. Note: The numbers 1–9 pertain to lamination stacking sequence code (see Table 1).

**Figure 4 polymers-13-02719-f004:**
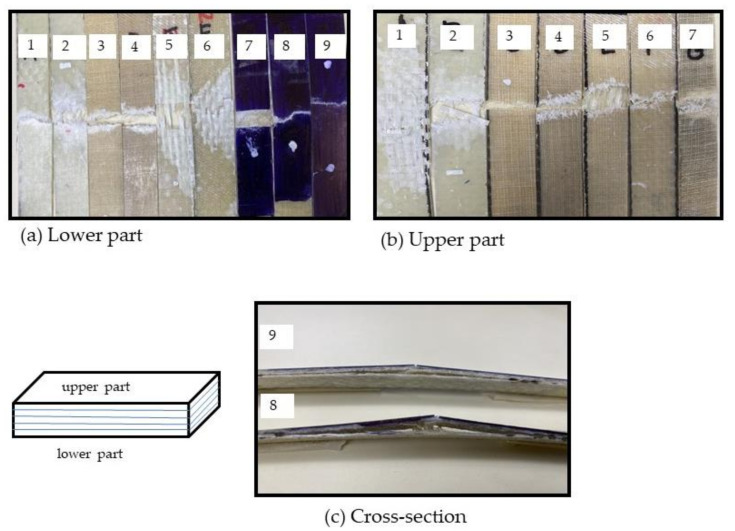
Digital images of the tensile fractures of the (**a**) lower part, (**b**) upper part, and (**c**) cross section of the laminated samples.

**Figure 5 polymers-13-02719-f005:**
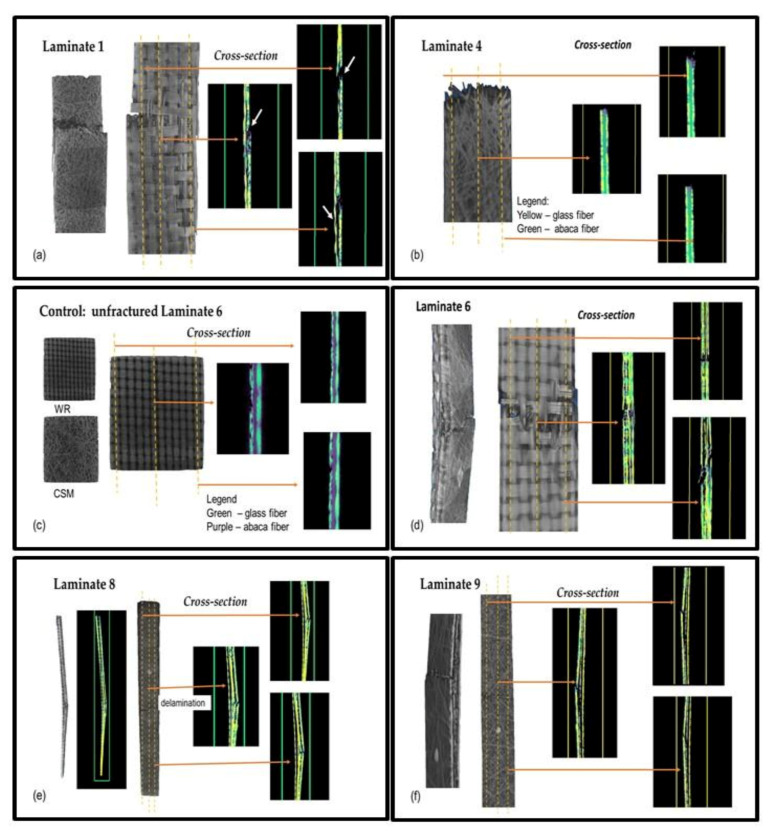
X-ray Computed Tomography images of the cross section of tensile fractures of the laminated samples (**a**) laminate 1, (**b**) laminate 4, (**c**) control-unfractured laminate 6, (**d**) laminate 6, (**e**) laminate 8, and (**f**) laminate 9.

**Figure 6 polymers-13-02719-f006:**
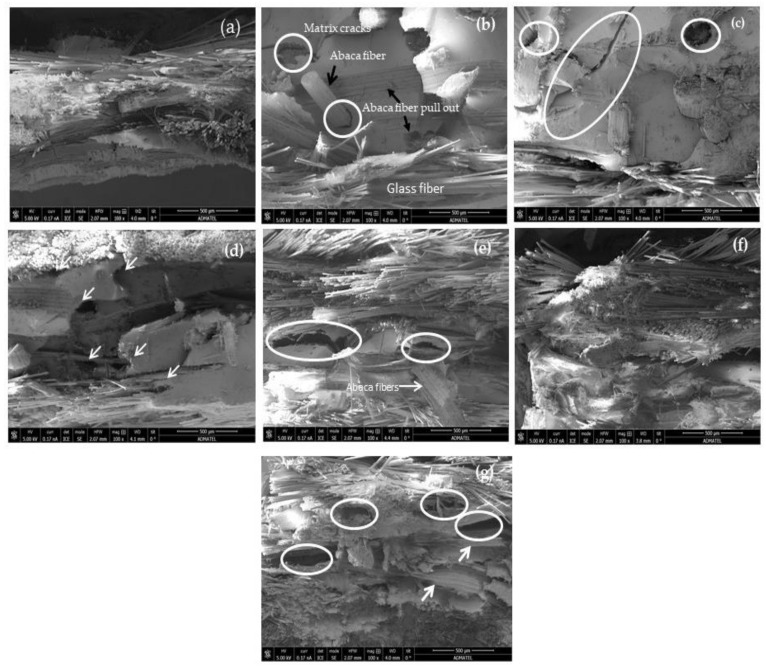
SEM images of tensile fractures of (**a**) laminate 1, (**b**) laminate 3, (**c**) laminate 4, (**d**) laminate 5, (**e**) laminate 6, (**f**) laminate 8, and (**g**) laminate 9. Note: Arrow indicates poor interaction between fibers and matrix, voids and matrix cracking.

**Table 1 polymers-13-02719-t001:** Lamination stacking sequence.

Lamination Stacking Sequence Code	Stacking Sequence	% V_AF_ (Abaca Fiber)	%V_GF_ (Glass Fiber)	Thickness (mm)
1	(CSM450)(CSM300) (WR600)	0	53	1.50
2	(CSM450)(WR600) (CSM450)	0	62	1.68
3	(CSM450)(abaca)_7_	21	12	2.27
4	(CSM450)(abaca)_7_ (CSM300)	18	19	2.67
5	(WR600)(abaca)_7_ (CSM300)	17	18	2.58
6	(WR600)(abaca)_3_(CSM300)	10	18	1.81
7	(CSM300)(CSM450)(abaca)_3_ (WR600)(abaca)	8	29	3.1
8	(CSM300)(CSM450)(abaca)_2_ (WR600)(abaca)	7	31	2.95
9	(CSM450)(abaca)_3_ (CSM300)(abaca)(WR600)(CSM300)	8	40	3.15

## Data Availability

The data presented in this study are available upon request from the corresponding author.

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
