# Peer review of "Mechanical Properties of Abaca–Glass Fiber Composites Fabricated by Vacuum-Assisted Resin Transfer Method"

_polymers, 2021, doi:10.3390/polym13162719_

Round 1

Reviewer 1 Report

The article "Mechanical Properties of Abaca-Glass fiber Composites Fabricated by Vacuum Assisted Resin Transfer Method" is interesting and up-to-date.

The introduction reflects the subject of the article well. I have comments on the description of the materials used in the study. The authors should provide the basic characteristics of abaca fibers, glass fibers and the polyester used. In line 81 there should be CSM 300 and CSM 450 instead of GSM 300 and GSM 400. In the description of the test methods in the characteristics of the Instron testing machine, the load cell value was given - 5 kg. I should be probably 5000 kg. In the discussion of the results, I miss a short summary and indication of individual hybrid composites with the best properties, in each of the chapters of mechanical properties testing. For example: tensile strength - composites 5 and 6; tensile modulus – composites 4 and 5. Flexural strength: composites 6 and 8; Flexural modulus 5 and 6. Impact strength 5, 6 and 8. On the base of such analysis the best hybrid composites should be indicated. For example: the best composite –no  5; second - composite no 6 and then composite no 8. In the description of impact tests in line 217 it should be: “The laminates coded 7 and 9.....”  instead of “The laminates 8 and 9…..”. There is also a minor mistake in line 241 and instead of: “Figures a, b & a” there should be “Figures a, b & c.”

After discussing the results of mechanical hybrid composites, authors should apply the failure morphology analysis and once again try to indicate the best of the tested hybrid composites for possible industrial applications. In my opinion the authors should also expand a bit the conclusions from the research with the description of all the composites made and indicate directions of their possible applications.

The literature of the article is extensive, but there are few articles from recent years (2 from 2020 and 1 from 2019). You could extend it by two or three more items from 2020-2021 (e.g. from the Journal of Natural Fiber).

Summarizing my review, the article is worth publishing in the Polymers journal with minor corrections and indicated additions.

Reviewer 2 Report

Marissa A. Paglicawan, Carlo S. Emolaga, Johanna Marie B. Sudayon, and Kenneth B. Tria

Mechanical Properties of Abaca-Glass fiber Composites Fabricated by Vacuum Assisted Resin Transfer Method

Abaca/glass fiber composites were prepared by vacuum-assisted resin transfer method (VARTM) and characterized. The use of natural fibers, like abaca fiber with glass fibers resulting hybrid composites may have some useful properties for some industrial applications. Characterization of the samples were done by modern methods. They used modern methods: tensile tests, flexural tests, Izod impact tests, computed x-ray tomography and scanning electron microscopy.

The meaning of the abbreviations CSM and GSM is not specified see line 81. (CSM = Chopped Strand Mat and GSM= Glass Strand Mat?)

Line 86: “at as shorter” instead of “at a shorter”

Table 1.: The unit of thickness (mm) is missing.

Table 1.: CSM450 is written, but in Materials line 81 shows GSM400.

Line 109 and 112: 50x25x2.5mm3, 63.5x12.7x3mm3, it would be better: 50mmx25mmx2.5mm, 63.5mmx12.7mmx3mm.

Figure 1. and Figure 2.: The scale of the x-coordinate axes must be made uniform.

Line 285: Chapter 5 is missing.

Changing the composition of the samples is not systematic enough, the number of samples is small.
